biomedical engineering/biomaterials

silk fibroin, adipose-derived stem cells, angiogenesis, angioirritation, chick chorioallantoic membrane, scaffolds

**Authors for correspondence:**
Weerapong Prasongchean
e-mail: Weerapong.p@chula.ac.th
Peerapat Thongnuek
e-mail: peerapat.t@chula.ac.th

# Angiogenic property of silk fibroin scaffolds with adipose-derived stem cells on chick chorioallantoic membrane

Tanapong Watchararot[1], Weerapong Prasongchean[1] and Peerapat Thongnuek[2,3,4]

[1]Department of Biochemistry and Microbiology, Faculty of Pharmaceutical Science, [2]Biomedical Engineering Program, Faculty of Engineering, [3]Biomaterial Engineering for Medical and Health Research Unit, Faculty of Engineering, and [4]Biomedical Engineering Research Center, Faculty of Engineering, Chulalongkorn University, Bangkok 10330, Thailand

(iD) PT, 0000-0002-9074-784X

Angiogenesis is a crucial step in tissue regeneration and repair. Biomaterials that allow or promote angiogenesis are thus beneficial. In this study, angiogenic properties of salt-leached silk fibroin (SF) scaffolds seeded with human adipose stem cells (hADSCs) were studied using chick chorioallantoic membrane (CAM) as a model. The hADSC-seeded SF scaffolds (SF-hADSC) with the porosity of $77.34 \pm 6.96\%$ and the pore diameter of $513.95 \pm 4.99\,\mu m$ were implanted on the CAM of chick embryos that were on an embryonic day 8 (E8) of development. The SF-hADSC scaffolds induced a spoke-wheel pattern of capillary network indicative of angiogenesis, which was evident since E11. Moreover, the ingrowth of blood vessels into the scaffolds was seen in histological sections. The unseeded scaffolds induced the same extent of angiogenesis later on E14. By contrast, the control group could not induce the same extent of angiogenesis. *In vitro* cytotoxicity tests and *in vivo* angioirritative study reaffirmed the biocompatibility of the scaffolds. This work highlighted that the biocompatible SF-hADSC scaffolds accelerate angiogenesis, and hence they can be a promising biomaterial for the regeneration of tissues that require angiogenesis.

## 1. Background

Angiogenesis is needed for the regeneration of various tissues [1]. Two types of angiogenesis are known: sprouting and

intussusceptive angiogenesis. Sprouting angiogenesis starts from vasodilation, degradation of the basement membrane, proliferation and migration of endothelial cells (ECs) from existing blood vessels [2]. The interplay between those processes and inflammation or hypoxia in wounded tissue generates pro-angiogenic factors that promote active proliferation and chemotactic migration of ECs toward the wounded area [1–3]. This results in the formation of new blood vessels. Intussusceptive angiogenesis involves the formation of transvascular tissue pillars that split the existing vessel [4]. This type of angiogenesis relies mainly on EC reorganization rather than proliferation [4].

Tissue regeneration is slowed down when angiogenesis is interrupted. For example, lines of evidence showed that inhibition of angiogenesis leads to retarded bone regeneration [5,6]. Indeed, the lack of vascular endothelial growth factor (VEGF), which results in impaired angiogenesis, leads to halted osteogenesis [7,8]. Angiogenesis is also essential for skin wound healing. The early stage of adult skin wound healing involves the formation of an extensive but rather disorganized capillary bed via angiogenesis [9].

Attempts have been made to accelerate angiogenesis by using scaffolding materials to ameliorate the tissue regeneration. For example, scaffolds seeded with bone marrow stromal cells that had been genetically engineered to deliver a pro-angiogenic growth factor showed enhanced angiogenesis in mice [10]. This then led to enhanced osteogenesis similarly to other reports with different scaffolds [5,11].

Natural polymers such as collagens, gelatin, silk fibroin and chitosan can be used to fabricate scaffolds, and they have drawn attention because of their biocompatibility, biodegradability and decent interaction with extracellular matrices [12,13]. Among those, silk fibroin (SF) produced from *Bombyx mori* silkworms exhibits relatively decent mechanical robustness, making it suitable for orthopaedic applications [14–16]. Many reports have shown the osteogenic potential of SF-based scaffolds [14,17–23]. However, the angiogenic potential, which is beneficial for osteogenesis, of SF is still unclear.

The angiogenic study can be done in the chorioallantoic membrane (CAM) of chick embryos [24,25]. The CAM is formed by the fusion of the outer mesodermal layer of the allantois and the mesodermal layer of the chorion during embryonic day 4 (E4) and E5 of development. Then, the network of blood vessels is gradually formed between the two layers. The central part of the CAM is fully developed approximately by E8–E10 [25]. At this time point, the CAM can support the weight of the scaffold materials [25]. The materials may be able to induce the spoke-wheel pattern formation representative of angiogenesis [26,27]. The CAM model provides many advantages for assessing the angiogenic potential, including less ethical concern, less cost and short developmental time. The model is also immunoincompetent until E17, providing an opportunity to bypass immunological reactions [25]. This allows angiogenic testing of the scaffolds fabricated from xenogenic protein [25]. Moreover, it allows live observation of angiogenesis, as the scaffold implanted on the CAM can be seen through an eggshell window.

In this study, we set to investigate the angiogenic potential of the SF scaffolds and the SF scaffolds seeded with hADSCs using the CAM as a model. We further exploited the CAM to study the biocompatibility *in vivo* using an angioirritative assay. This work demonstrated the angiogenic potential of hADSC-seeded SF scaffolds, and should also draw attention to the benefit of the CAM model in the tissue engineering field.

# 2. Methods

## 2.1. Egg preparation

Fertilized eggs were obtained from Kasetsart University, Thailand. They were incubated in an incubator for 3 days at 38.5°C. The eggs were windowed on their eggshell, and 2–3 ml of albumin were removed with a syringe. The window on the eggshell was closed with the Scotch tape. All steps were performed with aseptic techniques. All animal procedures were approved by Chulalongkorn University Animal Care and Use Protocol (CU-ACUP), Faculty of Pharmaceutical Sciences, Chulalongkorn University (Protocol no. 1633008).

## 2.2. Cell culture

The hADSCs were obtained from Krissanapong Manotham, MD after lipoaspiration. The protocol was approved by the Ethics Committee of Lerdsin General Hospital, Bangkok, Thailand. The cells were cultured in Dulbecco's modified Eagle medium with high glucose (DMEM; Cat No: 12100046; Gibco, USA) supplemented with 10% fetal bovine serum (FBS; Gibco, USA) and 1% penicillin and streptomycin (Gibco, USA), and incubated at 37°C, 5% $CO_2$ and 95% humidity. The medium was changed twice a week. For harvesting viable human adipose stem cells (hADSCs), we used the standard trypsinization procedure.

## 2.3. Silk fibroin solution preparation

Silk fibroin solution was prepared from the cocoons of Thai silkworms *B. mori* (kindly provided by Queen Sirikit Sericulture Center, Nakhonratchasima province, Thailand). Briefly, the cocoons were degummed by boiling for 20 min in an aqueous solution of $0.02\,M$ $Na_2CO_3$ (Sigma-Aldrich, Singapore). The resulting silk fibres were retrieved from the solution, and washed three times with deionized water. The degumming process was repeated once more. The silk fibres were air-dried. The dried fibres were then dissolved in $9.3\,M$ LiBr solution (Sigma-Aldrich, Singapore) for 4 h at 60°C. Next, the solution containing SF was dialysed for 3 days against deionized water using dialysis tubing (MWCO 12 000–16 000; Viskase Company Inc, Japan). The dialysate was centrifuged, and the supernatant was used for the scaffold fabrication.

## 2.4. Fabrication of salt-leached SF scaffolds

The porous scaffolds were prepared using NaCl as a porogen [28]. Briefly, 7 g of NaCl was slowly sprinkled into the SF solution in a cylindrical mould. The salt was removed by gentle stirring for 3 days in deionized water. The scaffolds were then dried and cut into the dimension of $5 \times 5 \times 2\,mm$. They were sterilized overnight under UV light in a biosafety cabinet before use. The scaffold porosity was determined using the liquid displacement technique according to the published research [22]. Hexane was used as the displacement liquid, as it permeates the SF scaffolds without swelling or shrinking the matrix. Each scaffold was submerged for 5 min in 1 ml *n*-hexane (the weight of 1 ml *n*-hexane = $W_1$). The hexane with the submerged scaffold was weighed ($W_2$). The scaffold was then removed, and the remaining hexane was weighed ($W_3$). The porosity of SF scaffolds was calculated using the following equation:

$$\% \text{ Porosity} = \left(\frac{W_1 - W_3}{W_2 - W_3}\right) \times 100,$$

where $W_1$ is the initial *n*-hexane weight, $W_2$ is the weight of *n*-hexane with the submerged scaffold and $W_3$ is the weight of the remaining *n*-hexane after removing the scaffold ($n = 5$).

## 2.5. Scanning electron microscope

The SF scaffolds were visualized by scanning electron microscope (SEM) (JSM-IT500HR, JEOL, USA). The scaffolds were glued on a copper plate and sputtered with gold before SEM imaging. The SEM was operated at the accelerating voltage of 3 keV. The pore diameters of the scaffolds were measured using ImageJ software.

## 2.6. hADSC seeding in the SF scaffolds

The cell seeding method was adopted from the literature [29]. The SF scaffolds were pre-wet in the culture media in 24-well plate; $1 \times 10^5$ hADSCs cells were seeded on the SF scaffolds. The cell-seeded scaffolds were incubated for 3 days at 37°C. The cell culture medium used in the experiments was Dulbecco's modified Eagle medium with high glucose (DMEM; Cat no: 12100046; Gibco, USA) supplemented with 10% (v/v) fetal bovine serum (FBS; Gibco, USA), 100 U ml$^{-1}$ penicillin (Gibco, USA) and 10 µg ml$^{-1}$ streptomycin (Gibco, USA). The cell-seeded scaffolds were incubated at 37°C, 5% $CO_2$ and 95% humidity.

## 2.7. MTT assay

After 3 days of cell seeding into the scaffolds, the culture media was removed. In total, 200 µl of MTT (Sigma-Aldrich, Singapore) was added to the scaffolds and incubated for 3.5 h at 37°C. The liquid was then carefully removed, and 1400 µl of DMSO (Merck, Germany) was added, followed by 15 min agitation. The optical absorbance was measured using a microplate reader (CALIOstar, Singapore) at the wavelength of 570 nm [30]. Four independent assays were performed.

## 2.8. Fluorescent staining

The method was modified from the literature [31]. After 3 days of cell seeding into the scaffolds, the cell-seeded scaffolds were gently washed with PBS. The scaffolds were then stained with 1 µg µl$^{-1}$ Hoechst 33342 (Thermo Scientific, USA) and 0.67 mM acridine orange (Sigma-Aldrich, USA). They were incubated

for 3 h at 37°C, and washed again to remove non-specific binding. The stained scaffolds were observed under an inverted fluorescence microscope (Olympus Ix51, Japan). The experiment was done in triplicate.

## 2.9. CAM assay

The scaffolds were placed on the CAM on E8. Filter paper (5.5 mm in diameter) was used in this experiment as a sham control. The embryos were monitored every 24 h. Images of the scaffolds on the CAM were taken through the window on the eggshell. The angiogenesis was analysed using a semi-quantitative method [32]. A score ranging from 0 to 5 was given to reflect the extent to which blood vessels formed and branched. Zero means no change in the vascular network while 1, 2 and 3 indicate a gradual increase in the number of new blood vessels; 4 and 5 mean extensive blood vessel density and branching of the new vessels [32]. Each group had five independent samples.

## 2.10. HET-CAM assay

The assay was modified from Wutzler *et al.* [33] to test for possible irritation to the blood vessels. The angioirritative effect of the SF scaffolds was compared against that of 0.1 M NaOH as a positive control and normal saline (0.9% NaCl) as a negative control. The eggs were incubated for 10 days, and windowed. The SF scaffolds or 100 µl of either NaOH or normal saline was applied on the CAM. Vessel haemorrhage, vessel lysis and coagulation were recorded under a stereomicroscope. An angioirritative grade was given to each image. The grade was calculated from the irritation score (IS). IS calculation was as follows:

$$\text{Irritation score (IS)} = \left(\frac{301 - t(h)}{300}\right) \times 5 + \left(\frac{301 - t(l)}{300}\right) \times 7 + \left(\frac{301 - t(c)}{300}\right) \times 9.$$

The time $t$ in seconds shows the onset of the following effects: $h$ = vascular haemorrhage, $l$ = vascular lysis, $c$ = coagulation. The calculated value of IS ranged between 0 and 21. The IS value was then converted into the angioirritative grade using the following criterion [34]:

0 = no irritation (IS = 0.00–0.90)
1 = slight irritation (IS = 1.0–4.9)
2 = moderate irritation (IS = 5.0–8.9)
3 = strong irritation (IS = 9.0–21.0).

The assay was done in triplicate.

## 2.11. Histological assessment

The implanted scaffolds with the CAM were fixed for 3 h with 4% paraformaldehyde (Merck, Germany) at room temperature. They were then cut, and placed in 30% sucrose solution for 48 h at 4°C. The fixed SF scaffolds with the CAM were processed with cryostat sectioning (Leica CM 1800, Germany). The thickness of each section was 40 µm. The samples were stained with haematoxylin and eosin (H&E). The sections were observed under a microscope.

## 2.12. Statistical analysis

The differences between the means in the MTT assay and the angiogenesis-score analysis were assessed using one-way ANOVA with Tukey's multiple comparison post-test. The $p$-value smaller than 0.05 was considered statistically significant. All statistical analyses were performed using IBM SPSS statistics 22 (Concurrent user license, license manager name 'splm.it.chula.ac.th', IBM).

# 3. Results

## 3.1. Physical characteristics of the salt-leached SF scaffolds

The structure of the SF scaffolds made from the NaCl salt-leaching method was analysed using SEM. We found that the SF scaffolds were highly porous, and the pores were interconnected (figure 1). Image analysis revealed that there was variation in the pore size with the average pore diameter of

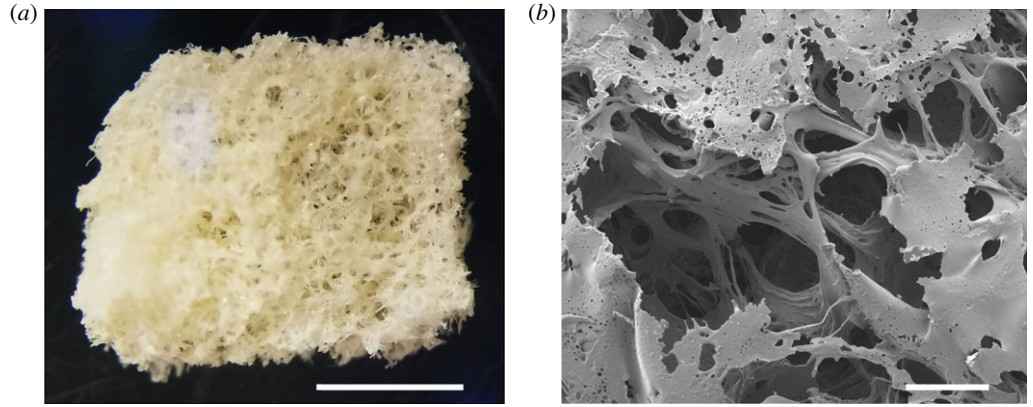

**Figure 1.** The macro- and microstructure of the SF scaffolds. (*a*) The SF scaffolds with the dimension of $5 \times 5 \times 2$ mm before implanting on the CAM. The scale bar, 2 mm. (*b*) The representative scanning electron micrograph of the SF scaffolds shows porous structures with pore interconnections. The scale bar, 200 μm.

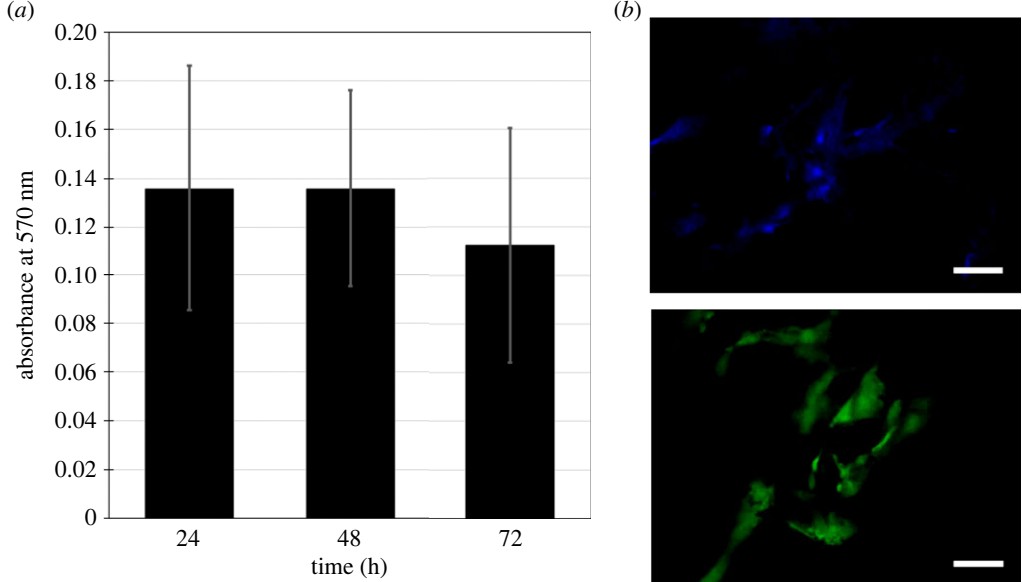

**Figure 2.** The viability of the hADSCs in the SF scaffolds by the MTT assay and fluorescence staining. (*a*) Absorbance at 570 nm in the MTT assay of the hADSC cells seeded in the SF scaffolds over 72 h ($n = 4$). The ANOVA test results in the *p*-value of 0.3134, meaning no significant difference. Error bars show s.d. (*b*) Representative Hoechst-acridine orange staining of hADSCs after 3-day incubation in the SF scaffolds ($n = 3$). Top: Hoechst staining; bottom: acridine orange staining. The scale bars, 500 μm.

$513.95 \pm 4.99$ μm. We also evaluated the porosity of the scaffolds using the liquid substitution method, and found that the average porosity was $77.34 \pm 6.96\%$.

## 3.2. hADSC survival in the SF scaffolds

The viability of hADSCs seeded in the scaffold was examined using the MTT assay and fluorescence staining. The relative absorbance in the MTT assay showed no significant difference after 3 days of cell culture (figure 2*a*). This means that there was still some cellular activity in the seeded scaffolds, suggesting that the hADSCs were still alive in the scaffolds for at least 3 days.

Moreover, the MTT assay result was supported by acridine orange and Hoechst staining. The acridine orange staining revealed the cells with a spindle shape reminiscent of the hADSC morphology (figure 2*b*, bottom). The Hoechst staining showed positions of the nuclei on the scaffolds (figure 2*b*, top). These altogether suggest that the SF scaffolds were non-cytotoxic to the hADCSs, allowing the cells to survive in the scaffolds for at least 3 days. Thus, the scaffolds seeded with the hADSCs were justified for the use in the CAM assay and others.

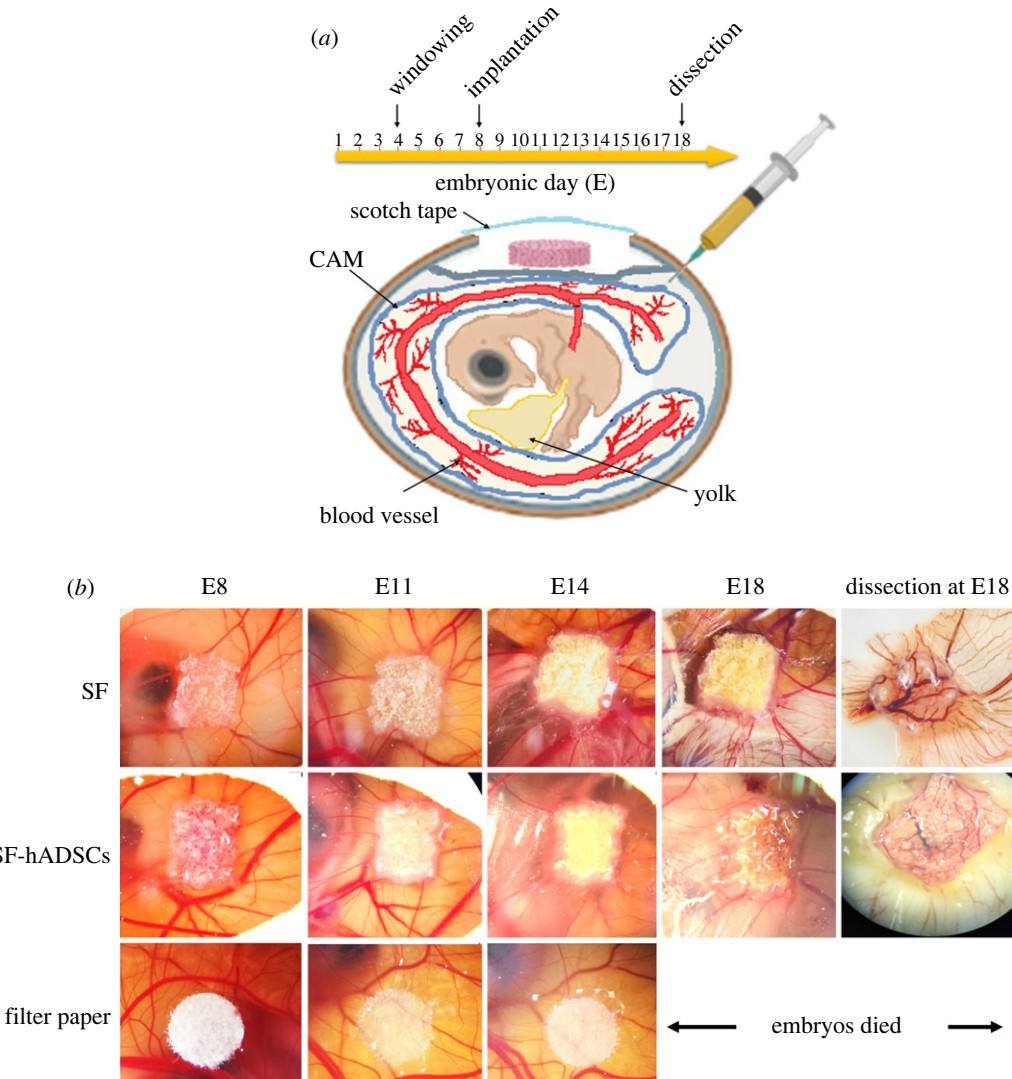

**Figure 3.** The angiogenic property of the scaffolds. (*a*) The CAM assay experimental set-up. (*b*) The spoke-wheel pattern of angiogenesis developed over the course of the embryonic development. The SF or SF-hADSC scaffolds were placed on the CAM of E8 embryos. The embryos with the scaffolds developed until E18. On E18, the scaffolds and the associated tissue were dissected to show vasculature underneath. The filter paper was used as a control. All embryos with the filter paper died before reaching E18 ($n = 5$).

## 3.3. The effect of the SF-hADSC scaffolds on angiogenesis

The spoke-wheel formation of the blood vessels around the scaffolds is indicative of the angiogenic property. We compared the onset and the extent to which the spoke wheel formed using a semi-quantitative scoring system (a score of 5 means substantial angiogenic property). The SF scaffolds induced the spoke-wheel pattern formation on E14, whereas the SF-hADSC scaffolds induced the spoke-wheel formation earlier on E11 (figure 3 and table 1). Indeed, the blood vessel density resulting from the SF-hADSC scaffolds on E11 was greater than the cell-free SF scaffolds on the same embryonic day. Later on E14, the density of vasculature in the SF group caught up with that in the SF-hADSC group. This finding indicated that the scaffolds either with or without cells could activate the angiogenesis, but the SF-hADSC scaffolds had greater angiogenic potential than the unseeded SF scaffolds. Moreover, we compared the angiogenic property between the filter paper as a control and the SF scaffolds. The result showed that the filter paper did not induce much change in the vasculature, and hence showed an angiogenesis score clearly smaller than that of the SF or SF-hADSC scaffolds. It is also worth noticing that a part of the CAM started to cover the edge of the SF-hADSC scaffolds at E14.

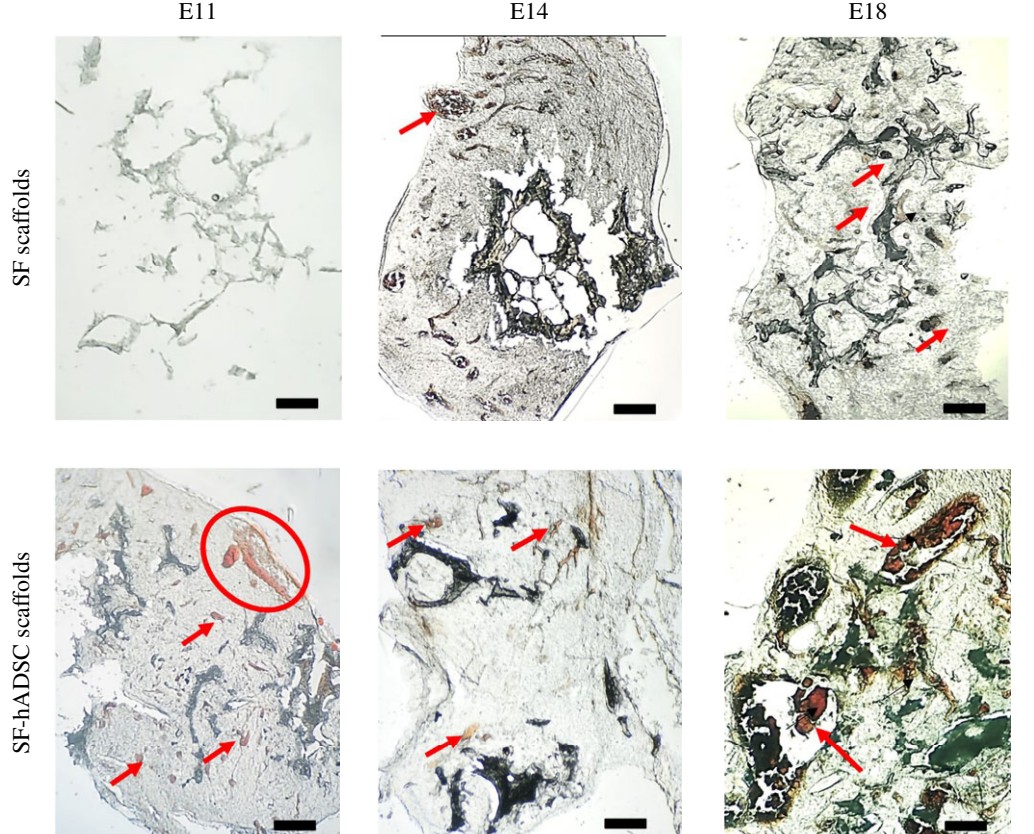

**Figure 4.** Histological analysis of the implanted SF-hADSC scaffolds compared with the SF scaffolds on different embryonic days. The representative images are shown. The sections were 40 µm thick. The red arrows indicate the blood vessels, and the red circle indicates a group of the blood vessels. The scale bar, 100 µm.

**Table 1.** Angiogenesis scoring. The values are mean score ± s.d. ($n = 5$). The superscripted letters indicate distinct statistical groups. The groups with different letters have significantly different means.

| scaffolds | angiogenesis score (0–5) | | |
| --- | --- | --- | --- |
| | E11 | E14 | E18 |
| SF | 2.40 ± 0.55[a] | 3.60 ± 0.55[d] | 3.75 ± 0.50[d] |
| SF-hADSC | 3.20 ± 0.45[b] | 3.80 ± 0.45[d] | 4.00 ± 0.00[d] |
| filter paper | 0.00 ± 0.00[c] | 2.33 ± 0.52[a] | embryos died |

## 3.4. Penetration of the blood vessels into the scaffolds

The implanted scaffolds were retrieved at different developmental time points, and dissected to visualize whether the blood vessels penetrated the porous structures in the scaffolds. We detected the blood vessels in both SF and SF-hADSC scaffolds (figure 4). Histological analysis revealed that the blood vessels penetrated throughout the scaffolds. Moreover, there were different sizes of the blood vessels inside the scaffolds. The penetration of the blood vessels was seen as early as at E11 when the SF-hADSC scaffolds were implanted. The SF scaffolds without hADSCs had the penetration starting from E14. This supports the mentioned spoke-wheel pattern analysis.

## 3.5. Vascular toxicity of the scaffolds

The angioirritative effect of the SF scaffolds was not reported elsewhere. Therefore, we studied the angioirritation of the SF scaffolds by the HET-CAM assay. The result showed no sign of blood vessel

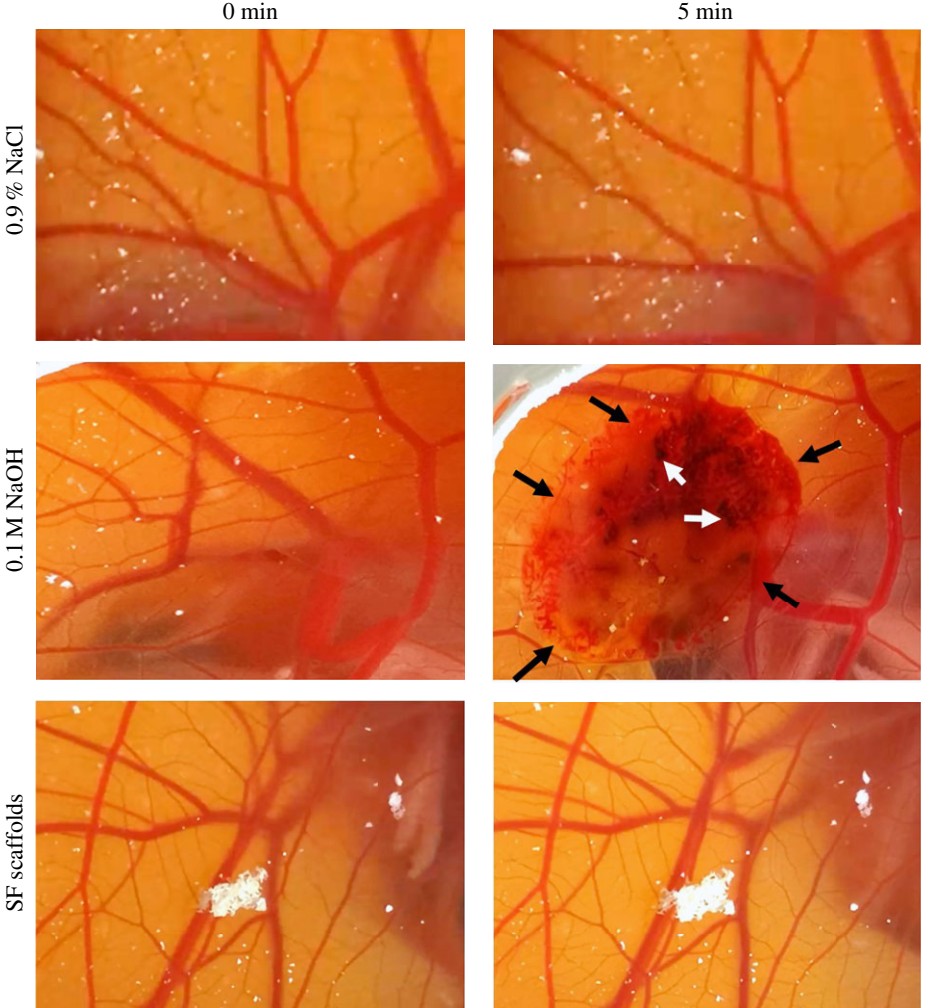

**Figure 5.** The angioirritation test of the SF scaffolds. The representative images are shown. The black arrows indicate the area of haemorrhage and vessel lysis. The white arrows indicate coagulation. The negative control that had been treated with the isotonic solution of 0.9% NaCl and the group that had been implanted with SF scaffolds received the IS score of 0. The positive control treated with 0.1 M NaOH resulted in the IS score of 3 ($n = 3$; electronic supplementary material, videos).

bleeding, vessel lysis or haemorrhage when the CAM had been in contact with the scaffolds, as the irritative score (IS) was 0 (figure 5). By contrast, the haemorrhage could clearly be seen when 0.1 M NaOH was given as a positive control, and thus the IS score was 3 (maximum). This means that the SF scaffolds had no agioirritative effect.

# 4. Discussion

The CAM model has been overlooked in the field of tissue engineering even though it has many advantages, e.g. less ethical concern, less cost, short developmental time and easy to perform. Chick embryos lack a complete immune system, so they act as an *in vivo* system for evaluating the efficacy of grafts without rejection [24,25,27]. The developmental time frame of chick embryos is significantly shorter than that of other animal models such as rabbits. The chick embryo development only takes 21 days before hatching, whereas rabbits take 31 days [35,36]. This will surely speed up experiments. Moreover, the experimental set-up makes angiogenic observation simple. A window in the eggshell can be made for live observation without sacrificing the animals [24,37]. Therefore, we selected the chick CAM as a model to investigate the angiogenic potential of our biocompatible SF scaffolds with/without hADSCs.

We demonstrated the *in vitro* and *in vivo* biocompatibility of the SF scaffolds. Although the biocompatibility of SF has long been proved, we showed the angiocompatibility of SF scaffolds for the first time using the

HET-CAM assay. This ensures that the leach-out chemicals or the degradation products of the scaffolds do not cause angioirritation that results in haemorrhage, vascular lysis or coagulation.

In the CAM assay, we found that the SF scaffolds even without the hADSCs could promote the angiogenesis. Indeed, the spoke-wheel formation was observed 6 days after implantation, whereas the filter paper, as a sham control, could not promote the angiogenesis. Noteworthy, we used the filter paper to prove that not any kind of material can promote the angiogenesis when placed on the CAM. We believe that the filter paper is biologically inert, as they have been used as a negative control in various angiogenic molecule-delivery experiments [30,38,39]. Our finding is consistent with a previously published report [40]. In that study, several blood vessels were seen underneath the freeze-dried SF scaffolds after 4 days of implantation on the CAM [40]. Their finding also suggested the angiogenic capability of the SF scaffolds, although a sham control was not included. In addition, our study revealed the penetration of the blood vessels into the porous structures of the scaffolds. These altogether confirm the angiogenic potential of the SF scaffolds even without any cells seeded.

When the hADSCs were seeded into the SF scaffolds, they ameliorated the angiogenic potential of the scaffolds. We found that the SF-hADSC scaffolds induced the spoke-wheel formation earlier than the SF scaffolds alone. As shown in our result, the SF-hADSC scaffolds implanted onto the CAM of E8 embryos induced the formation of the blood vessels that became apparent on E11. This means that the SF-hADSC scaffolds took only 3 days while the SF scaffolds alone took 6 days. A previous study conducted using adenovirus vector-modified SF scaffolds (SF-Ad scaffolds) containing angiogenic genes showed that the SF-Ad scaffolds promoted the microvascular network formation 4 days after implantation onto E8 CAM [40]. Another study was done by implanting SF-monofilament of polyethylene terephthalate (SF-PET) and monofilament of polyethylene terephthalate (PET) textile scaffolds in E10 embryos [41]. SF-PET and PET were well integrated into the CAM after 4 days of implantation. The scaffolds made from PET alone resulted in less blood vessel density compared to the SF-PET scaffolds after 4 days of incubation [41]. Both mentioned works reported the angiogenic induction by the modified SF scaffolds after 4 days of implantation on the CAM while our SF-hADSC scaffolds induced the angiogenesis within 3 days after implantation [40,41]. This means that our SF scaffolds with the hADSCs promote the angiogenesis earlier than other modifications.

It is worth mentioning that selecting embryonic days for implantation may affect the rate of angiogenesis. During chick development, the blood vessels start to develop on E4, and continue to grow exponentially until E14 to serve as a respiratory organ [37]. Initially, the formation of the CAM resulted in the development of the vascularized mesoderm composed of arteries, veins and an intricate capillary plexus. Then the capillaries proliferate until E11 [42]. We decided to select E8 as the implantation starting point because the development becomes slower in the later stages. Furthermore, chick embryos lack the specific immune system until E14 [43]. We believe that implanting on E8 was early enough to observe changes induced by the scaffolds. The presence of the specific immune system after E14 might explain why the embryos implanted with the filter paper died before reaching E18.

In addition, the fate of the hADSCs in the scaffolds should be studied in the future. Our experiment provides preliminary data that the presence of the stem cells in the SF scaffolds accelerates the angiogenesis, but cannot pinpoint whether the new vessels were formed from the hADSCs themselves or from the interaction between the hADSCs and the cells of the chick embryo.

The mechanism of blood vessel penetration is still unclear. We hypothesized that the possible mechanism for the angiogenesis could be due to the contribution from both SF scaffolds and the hADSCs. The scaffolds provide the surface area for cell adhesion, and their porosity allows adequate mass transfer for cells to maintain their lives. Moreover, the degradation products of the SF scaffolds may be beneficial for the angiogenesis. We showed by the fluorescence microscopy that the hADSCs still attached to the SF scaffolds after 3 days of seeding, and the hADSCs survived for at least 3 days as observed by the MTT assay. A group of researchers reported that cell adhesion to a surface is caused by the deposition and remodelling of extracellular matrices, which could modulate biochemical signalling [44]. We believe that the cell adhesion-mediated signalling may lead to the enhanced angiogenesis. Another work showed that the scaffolds made of SF alone allowed cell attachment even though such attachment could be improved by modifying the surface of the SF material [45]. The *in vitro* analysis in this work suggests that the hADSC cells in the implanted scaffolds were alive and active during 3 days of angiogenesis from E8 to E11.

In addition, the scaffold porosity can support cell proliferation, differentiation, aggregation and vascularization [46]. In our study, the SF scaffolds had the average pore size of $513.95 \pm 4.99\,\mu m$. It was established that the pore size between 250 and $500\,\mu m$ is optimum for promoting cell proliferation and ECM production [47,48].

Furthermore, the degradation products of the SF scaffolds may activate the angiogenesis. The SF scaffold degradation could be due to enzymatic proteolysis such as by chymotrypsin or carboxylase present in the CAM [49]. Once degraded, the amino acids are resorbed by the surrounding cells. Evidence from immunohistochemistry staining of the CD34 proliferation marker suggested that the SF scaffold-degradation products could promote EC proliferation [50]. The endothelial proliferation may then lead to the formation of new capillaries.

The contribution of the hADSCs can be due to the release of the angiogenic growth factors such as VEGF and FGF-2. There was evidence showing that the hADSCs could slowly release VEGF from hydrogel [29]. The same work documented the supporting role of the hADSCs when co-cultured with the SVEC4–10 vascular ECs [29].

Apart from the release of angiogenic factors, many reports demonstrated the positive interaction between the ADSCs and the ECs. A study conducted in nude mice revealed an enhanced blood vessel formation when human adipose mesenchymal stem cells and endothelial colony-forming cells (ECFCs) were co-cultured in the Matrigel, and injected subcutaneously [51]. Other evidence from the HUVEC-ADSC co-culture showed that the ADSCs promoted angiogenic sprouting. They demonstrated that the ADSCs released matrix metalloproteinase (MMP) to digest the collagen hydrogel, allowing endothelial sprouting [52]. The enzyme MMPs were also shown to digest the SF [53]. The ECM microarchitecture was shown to promote the endothelial sprouting [54]. These altogether may explain how the SF-hADSC scaffolds in our study promoted the angiogenesis.

# 5. Conclusion

This work demonstrated the angiogenic potential of the SF-hADSC scaffolds using the chick embryo CAM as a model. Our SF-hADSC scaffolds showed the angiogenic effect earlier than other SF scaffolds reported in the literature. We also showed that the SF scaffolds without the cells could promote the angiogenesis, but their angiogenic effect was seen later than that of the SF-hADSC scaffolds. The HET-CAM assay revealed no angioirritative effect of the SF scaffolds. Our work emphasizes the potential of the SF-based biomaterials for tissue regeneration, as the tissue regeneration typically requires angiogenesis.

Ethics. All animal procedures were approved by Chulalongkorn University Animal Care and Use Protocol (CU-ACUP), Faculty of Pharmaceutical Sciences, Chulalongkorn University (Protocol no. 1633008).

Data accessibility. Data available from the Dryad Digital Repository: https://doi.org/10.5061/dryad.8931zcrnz [55].

Authors' contributions. T.W. carried out the laboratory work, participated in data analysis, participated in the design of the study and drafted the manuscript; W.P. and P.T. participated in data analysis, participated in the design of the study, drafted the manuscript and found financial support for the project.

Competing interests. We declare we have no competing interests.

Funding. This Research was funded by Chulalongkorn University CU_GR_62_69_21_11. T.W. was supported by the 100th year Anniversary Chulalongkorn University Fund for Doctoral Scholarship.

Acknowledgements. The authors acknowledge assistance from Krissanapong Manotham, MD for hADSC collection.

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
