## [Peer Review File · Royal Society Open Science]

Review History

RSOS-201618.R0 (Original submission)

Review form: Reviewer 1

Is the manuscript scientifically sound in its present form?

No

Are the interpretations and conclusions justified by the results?

No

Is the language acceptable?

Yes

Do you have any ethical concerns with this paper?

No

Have you any concerns about statistical analyses in this paper?

Yes

Recommendation?

Major revision is needed (please make suggestions in comments)

Comments to the Author(s)

They premise of the article is very interesting, and the experimental setup is straightforward and easy to understand. However, several questions are raised with some of the Methods explained in the paper. The reviewer feels relevant information has either been withheld, or perhaps not investigated, and suggests that these should be included in the manuscript. In-depth comments can be found in the attached file (Appendix A).

Review form: Reviewer 2

Is the manuscript scientifically sound in its present form?

Yes

Are the interpretations and conclusions justified by the results?

Yes

Is the language acceptable?

Yes

Do you have any ethical concerns with this paper?

No

Have you any concerns about statistical analyses in this paper?

No

Recommendation?

Major revision is needed (please make suggestions in comments)

Comments to the Author(s)

The manuscript written by Tanapong et al demonstrated the angiogenic potential of silk fibroin scaffolds in the presence and absence of hADSC using CAM technique. The topic is of interest to the biomaterials and tissue engineering community. The results supported the claim made in this study. However, few points have to be addressed prior to publication.

Majors:

- On page 12 in line 38 – 43, the authors wrote “Moreover, there were different sizes of blood vessels inside the scaffolds. The penetration of the blood vessels was seen as early as at E11 when the SF-hADSC scaffolds were implanted.”. This information is descriptive rather than qualitative. This reviewer suggests the authors to quantify the blood vessel size and area of blood vessels per cross-sectional area of the scaffold. This is important for the study of angiogenic potential and adds up qualitative information to Figure 4.
- This reviewer would suggest the presentation of the data using mean +/- SD, instead of mean +/- SEM.
- For the angiogenic test, only IS score of 0 and 3 has been reported. How many independent experiments have been performed? This should be mentioned.
- Limitation of the study should be discussed. Especially, the fate of the hADSC in SF scaffolds during and upon the CAM assay was not studied.

- The authors should discuss why the filter paper in the CAM assay triggers embryonic death.

Minors:

- Glucose concentration of the DMEM media should be mention in the cell culture section in the materials and methods, since it is critical for culture of hADSC. The authors can also provide the catalog number of the media.
- In the cell scanning electron microscope section in the materials and methods, please provide the settings for the imaging.
- This reviewer would suggest to add “representative images of ...” in Figure 1B, 2B, 4 and 5.
- The number of experiments should be stated in Figure 2 and Table 1
- The author discussed on collagen hydrogel and angiogenesis on Page 17 line 37-43. Recent literature suggests that “collagen microarchitecture can modulate endothelial sprouting by modulating hADSC proangiogenic capability. (Seo B et al 2020 PNAS – DOI: 10.1073/pnas.1919394117)”. In addition, proinflammatory mediators, such as IL-6, could also trigger angiogenic potential in 3D scaffolds (Sapudom J et al 2020 Bioengineering, DOI 10.3390/bioengineering7020033) (Witzel I et al 2018 Adv Healthcare Matter. DOI 10.1002/adhm.201801126). This can be added to the discussion.

Decision letter (RSOS-201618.R0)

Dear Dr Thongnuek

The Editors assigned to your paper RSOS-201618 "Angiogenic property of silk fibroin scaffolds with adipose-derived stem cells on chick chorioallantoic membrane" have now received comments from reviewers and would like you to revise the paper in accordance with the reviewer comments and any comments from the Editors. Please note this decision does not guarantee eventual acceptance.

Please submit your revised manuscript and required files (see below) no later than 21 days from today's (ie 21-Jan-2021) date. Note: the ScholarOne system will 'lock' if submission of the revision is attempted 21 or more days after the deadline. If you do not think you will be able to meet this deadline please contact the editorial office immediately.

on behalf of R. Kerry Rowe (Subject Editor)
openscience@royalsociety.org

Associate Editor Comments to Author:

Comments to the Author:

The two reviewers have each identified a number of matters that need to be addressed in a revision and re-assessed with further reviewer attention. Reviewer 1 notes that the methods are 'thinner' than they would expect, so this will need addressing. The second reviewer has provided a number of useful comments, but also has recommended that the authors include several of their own works as citations - this 'citation stacking' should be discouraged. Only if the authors consider that these references do add value to their own paper should they include them - and the editors would expect the authors to address why they have included them in their rebuttal. While the editors recognise that reviewers will have published work that may be relevant, the point of acting as a peer reviewer is to support and encourage and improve submitted papers - it is manifestly not to support a reviewer's publication record.

Reviewer comments to Author:

Reviewer: 1

Comments to the Author(s)

They premise of the article is very interesting, and the experimental setup is straightforward and easy to understand. However, several questions are raised with some of the Methods explained in the paper. The reviewer feels relevant information has either been withheld, or perhaps not investigated, and suggests that these should be included in the manuscript. In-depth comments can be found in the attached file.

Reviewer: 2

Comments to the Author(s)

The manuscript written by Tanapong et al demonstrated the angiogenic potential of silk fibroin scaffolds in the presence and absence of hADSC using CAM technique. The topic is of interest to the biomaterials and tissue engineering community. The results supported the claim made in this study. However, few points have to be addressed prior to publication.

Majors:

- On page 12 in line 38 – 43, the authors wrote “Moreover, there were different sizes of blood vessels inside the scaffolds. The penetration of the blood vessels was seen as early as at E11 when the SF-hADSC scaffolds were implanted.”. This information is descriptive rather than qualitative. This reviewer suggests the authors to quantify the blood vessel size and area of blood vessels per cross-sectional area of the scaffold. This is important for the study of angiogenic potential and adds up qualitative information to Figure 4.
- This reviewer would suggest the presentation of the data using mean +/- SD, instead of mean +/- SEM.
- For the angi irritation test, only IS score of 0 and 3 has been reported. How many independent experiments have been performed? This should be mentioned.
- Limitation of the study should be discussed. Especially, the fate of the hADSC in SF scaffolds during and upon the CAM assay was not studied.
- The authors should discuss why the filter paper in the CAM assay triggers embryonic death.

Minors:

- Glucose concentration of the DMEM media should be mention in the cell culture section in the materials and methods, since it is critical for culture of hADSC. The authors can also provide the catalog number of the media.
- In the cell scanning electron microscope section in the materials and methods, please provide the settings for the imaging.
- This reviewer would suggest to add “representative images of ... ” in Figure 1B, 2B, 4 and 5.
- The number of experiments should be stated in Figure 2 and Table 1
- The author discussed on collagen hydrogel and angiogenesis on Page 17 line 37-43. Recent literature suggests that “collagen microarchitecture can modulate endothelial sprouting by modulating hADSC proangiogenic capability. (Seo B et al 2020 PNAS – DOI: 10.1073/pnas.1919394117)”. In addition, proinflammatory mediators, such as IL-6, could also trigger angiogenic potential in 3D scaffolds (Sapudom J et al 2020 Bioengineering. DOI 10.3390/bioengineering7020033) (Witzel I et al 2018 Adv Healthcare Matter. DOI 10.1002/adhm.201801126). This can be added to the discussion.

===PREPARING YOUR MANUSCRIPT===

While not essential, it will speed up the preparation of your manuscript proof if accepted if you format your references/bibliography in Vancouver style (please see

<https://royalsociety.org/journals/authors/author-guidelines/#formatting>). You should include DOIs for as many of the references as possible.

===PREPARING YOUR REVISION IN SCHOLARONE===

Author's Response to Decision Letter for (RSOS-201618.R0)

See Appendix B.

RSOS-201618.R1 (Revision)

Review form: Reviewer 1

Is the manuscript scientifically sound in its present form?

Yes

Are the interpretations and conclusions justified by the results?

Yes

Is the language acceptable?

Yes

Do you have any ethical concerns with this paper?

No

Have you any concerns about statistical analyses in this paper?

No

Recommendation?

Accept as is

Comments to the Author(s)

The authors replied to the comments in a satisfying way and adequate changes were made in the manuscript based on the comments of the previous review round.

Review form: Reviewer 2

Is the manuscript scientifically sound in its present form?

Yes

Are the interpretations and conclusions justified by the results?

Yes

Is the language acceptable?

Yes

Do you have any ethical concerns with this paper?

No

Have you any concerns about statistical analyses in this paper?

No

Recommendation?

Accept as is

Comments to the Author(s)

-

Decision letter (RSOS-201618.R1)

Dear Dr Thongnuek,

It is a pleasure to accept your manuscript entitled "Angiogenic property of silk fibroin scaffolds with adipose-derived stem cells on chick chorioallantoic membrane" in its current form for publication in Royal Society Open Science. The comments of the reviewer(s) who reviewed your manuscript are included at the foot of this letter.

Kind regards,

Anita Kristiansen
Editorial Coordinator

on behalf of R. Kerry Rowe (Subject Editor)
openscience@royalsociety.org

Associate Editor Comments to Author:

Comments to the Author:

The reviewers who have assessed your paper now recommend publication - congratulations! That said, the editor draws the readers attention to the fact that some of the additional experiments recommended by the reviewers could not be conducted, owing to the temporary closure of the research facility. Thus, while the editors are prepared to accept the paper is as ready for publication as practical, the authors are recommended to conduct the follow-up work when possible, and consider whether an update or comment will be needed at that time.

Reviewer comments to Author:

Reviewer: 1

Comments to the Author(s)

The authors replied to the comments in a satisfying way and adequate changes were made in the manuscript based on the comments of the previous review round.

Appendix A

Language

- 1) In general: English is understandable but definitely needs editing as many small mistakes are noticeable in the text (see a few examples below). Also, sometimes sentences are phrased oddly (see examples below), and should be edited as well.
 - a. Small mistakes are noticeable throughout the text:
 - i. Use 'the' in front of 'chick chorioallantoic membrane' or 'CAM model' etc.
 - ii. Page 3:
 1. 27: in THE formation
 2. 50: drawn attention\$
 - b. Phrasing is a bit odd in several sentences throughout the whole text
 - i. Page 4, 3: "A great amount of work have shown the osteogenic potential of silk fibroin-based scaffolds"
 - ii. Page 6, 41: "Four independent assays were done to find averages."
 - iii. Abstract: "In contrast, the control group could not induce angiogenesis to the same extent even as late."
 - iv. Etc.

Content

- 1) Page 3, 19-23: authors state that "Angiogenesis starts...existing blood vessels." This is not entirely correct as there are angiogenic processes (intussusceptive angiogenesis) that do not require these steps. Please rephrase.
- 2) Page 4, 16-18: authors state that "The central portion...scaffold materials." This is a confusing statement, please rephrase.
- 3) In the Methods sections, the text sometimes refers to other published work to explain some of their protocols, but without much further explanation, e.g. page 6, 5-6 : "displacement technique according to published research", and page 6, 22-23: "Cell seeding method was adopted from literature." For example, the exact content of the culture media referred to in the cell seeding method should be given.
- 4) It is not clear from the Methods whether the plain SF-scaffolds were also pre-wet in culture media but without any cell seeding. For optimal comparison, both SF – and SF-hADSC scaffolds should be treated exactly the same, with the only difference being the lack of cells in the SF-scaffolds. Based on figure 3B, it seems that the SF-hADSC scaffolds are wet but the plain SF-scaffolds are dry.
- 5) Page 7, 18-20: "0 means no change in vascular network while 1-5 means extensive blood vessel branching." Scoring system should be further explained as it seems now that there is hardly any difference between the scores 1-5.
- 6) Please mention in the Methods section which statistical test was used and when.
- 7) In the Results section, the authors state that they used filter paper as a control to analyze the angiogenic properties of the SF-scaffolds, but there is no mention of this in the Methods section. Then, it is unclear why exactly filter paper was used as sham control and not some other porous material, because blood vessels will never be able to grow into the filter paper. In addition, it is mentioned that 5 embryos were used for the filter paper experiments (only in the figure caption), but it is nowhere mentioned how many embryos were used to investigate the angiogenic properties of the SF-scaffold and SF-hADSC scaffold.
- 8) Histological sections were made with cryosectioning. However, it is known that the morphology of cryosections is far worse than that of regular paraffin-embedded samples.

This is reflected in Figure 4, where especially the E11 SF-scaffold picture is very unclear. Is there a reason why the samples were not processed for paraffin embedding as this would have less an impact on the morphology, thus lowering the chances that any small capillaries invading the SF-scaffolds would have been missed? Also, more and better (at least higher magnification) histological figures would be appreciated.

- 9) In figure 5, it seems that the 0 and 5 min pictures of 0.9% NaCl are the same, and the SF scaffolds pics are also highly similar. The reviewer comes to this conclusion as the positions of the blood vessels and embryo are identical, which is nearly impossible if pictures are taken with 5 minutes difference in between (own experience). Even if there is no angiirritation from the NaCl solution and the SF scaffolds, please provide the appropriate pictures.
- 10) As of now, the manuscript is mainly descriptive as no statistical analysis is performed to determine the difference between the angiogenic response between the two groups of SF-scaffolds on the different timestamps. Being merely descriptive is perfectly acceptable, but as a way to make the paper more scientifically sound, the reviewer suggests that statistical analysis is performed, and also that the authors include a 'positive' control within the groups to determine to what extent the hADSC are able to induce angiogenesis.

References

- 1) Reference nr 4 is incomplete

Appendix B

Reviewer 1

1) Page 3, 19-23: authors state that “Angiogenesis starts...existing blood vessels.” This is not entirely correct as there are angiogenic processes (intussusceptive angiogenesis) that do not require these steps. Please rephrase.

In that paragraph, we added “Two types of angiogenesis are known: sprouting and intussusceptive angiogenesis. Sprouting angiogenesis starts from..... existing blood vessels.”, and “Intussusceptive angiogenesis involves the formation of transvascular tissue pillars that split the existing vessel [4]. This type of angiogenesis relies mainly on EC reorganization rather than proliferation [4].”. Also, a reference (Ref no. 4) about the intussusceptive angiogenesis was added.”

2) Page 4, 16-18: authors state that “The central portion...scaffold materials.” This is a confusing statement, please rephrase.

We changed it to “The central part of the CAM is fully developed approximately by E8 to E10 [24]. At this time point, the CAM can support the weight of scaffold materials [24].”.

3) In the Methods sections, the text sometimes refers to other published work to explain some of their protocols, but without much further explanation, e.g. page 6, 5-6 : “displacement technique according to published research”, and page 6, 22-23: “Cell seeding method was adopted from literature.” For example, the exact content of the culture media referred to in the cell seeding method should be given.

For the porosity determination, we added “Hexane was used as the displacement liquid, as it permeates the SF scaffolds without swelling or shrinking the matrix. Each scaffold was submerged for 5 minutes in 1 ml n-hexane (the weight of 1ml n-hexane = W_1). The hexane with the submerged scaffold was weighed (W_2). The scaffold was then removed, and the remaining hexane was weighed (W_3). The porosity of SF scaffolds was calculated using the following equation.

$$\% \text{ Porosity} = \left(\frac{W_1 - W_3}{W_2 - W_3} \right) \times 100$$

When W_1 is the initial n-hexane weight, W_2 is the weight of n-hexane with the submerged scaffold, and W_3 is the weight of the remaining n-hexane after removing the scaffold. (n=5)”

For the cell seeding section, we added the media composition and cell culture condition; “The cell culture media used in the experiments was Dulbecco’s Modified Eagle Medium with high glucose (DMEM; Cat No: 12100046; Gibco, USA) supplemented with 10%(v/v) fetal bovine serum (FBS; Gibco, USA) and 100 U/ml penicillin (Gibco, USA) and 10 µg/ml streptomycin (Gibco, USA). The cell-seeded scaffolds were incubated at 37°C, 5% CO₂ and 95% humidity.”.

4) It is not clear from the Methods whether the plain SF-scaffolds were also pre-wet in culture media but without any cell seeding. For optimal comparison, both SF – and SF-hADSC scaffolds should be treated exactly the same, with the only difference being the lack of cells in the SF-scaffolds. Based on figure 3B, it seems that the SF-hADSC scaffolds are wet but the plain SF-scaffolds are dry.

We agree with the reviewer, and we thus changed the pictures. Actually, we tried both prewet and dry SF scaffolds. They gave indifferent results. We believe that this could be because the dry SF scaffolds immediately absorbed water and wet itself. The volume of water absorbed from the egg might be too small to cause any effect on angiogenesis. It is known that silk fibroin porous scaffold does not absorb much water compared to other biopolymers. Many documents report the water-swelling property of silk fibroin scaffolds, and they all pointed to the fact that the SF scaffolds do not absorb much water. We would like to provide examples of the reports:

- Grabska-Zielińska S, Sionkowska A, Reczyńska K, Pamuła E. Physico-Chemical Characterization and Biological Tests of Collagen/Silk Fibroin/Chitosan Scaffolds Cross-Linked by Dialdehyde Starch. *Polymers (Basel)*. 2020 Feb 7;12(2):372. doi: 10.3390/polym12020372. PMID: 32046018; PMCID: PMC7077405.
- Burette, F., Bouchard, F., Pellerin, C. *et al.* Cell-culture compatible silk fibroin scaffolds concomitantly patterned by freezing conditions and salt concentration. *Polym. Bull.* **67**, 159–175 (2011). <https://doi.org/10.1007/s00289-010-0438-z>

5) Page 7, 18-20: “0 means no change in vascular network while 1-5 means extensive blood vessel branching.” Scoring system should be further explained as it seems now that there is hardly any difference between the scores 1-5.

We added “0 means no change in the vascular network while 1, 2, and 3 indicate a gradual increase in the number of new blood vessels. 4 and 5 means extensive blood vessel density and branching of the new vessels [32]”. The scoring was done based on the reference given.

6) Please mention in the Methods section which statistical test was used and when.

We added in the statistical analysis subsection in the method section that “The differences between the means in the MTT assay and the angiogenesis-score analysis were assessed using one-way ANOVA with Tukey’s multiple comparison posttest. The p-value smaller than 0.05 was considered statistically significant.”

7) In the Results section, the authors state that they used filter paper as a control to analyze the angiogenic properties of the SF-scaffolds, but there is no mention of this in the Methods section. Then, it is unclear why exactly filter paper was used as sham control and not some other porous material, because blood vessels will never be able to grow into the filter paper. In addition, it is mentioned that 5 embryos were used for the filter paper experiments (only in the figure caption), but it is nowhere mentioned how many embryos were used to investigate the angiogenic properties of the SF-scaffold and SF-hADSC scaffold.

First, we added into the method section that we used filter paper as a sham control. “Filter paper (5.5 mm in diameter) was used in this experiment as a sham control.”.

Also, we added our reason to the discussion section. “Noteworthy, we used filter paper to prove that not any kind of material can promote angiogenesis when placed on CAM. We believe that the filter paper is biologically inert, as they have been used as a negative control in various angiogenic molecule-delivery experiments [38–40].”

We used 5 embryos for each group. We added to the CAM assay subsection in the method that “Each group had 5 independent samples.”

8) Histological sections were made with cryosectioning. However, it is known that the morphology of cryosections is far worse than that of regular paraffin-embedded samples. This is reflected in Figure 4, where especially the E11 SF-scaffold picture is very unclear. Is there a reason why the samples were not processed for paraffin embedding as this would have less an impact on the morphology, thus lowering the changes that any small capillaries invading the SF-scaffolds would have been missed? Also, more and better (at least higher magnification) histological figures would be appreciated.

We used cryosectioning mainly because it could be done quicker than paraffin-sectioning, and we did not intend to stain any molecule with antibody. We believe that the result in Figure 4 is consistent with that in Figure 3 and table 1. All of them suggests that the angiogenesis promoted by SF-hADSC started earlier than SF alone.

Sectioning of paraffin-embedded samples might have given better pictures. However, the chick embryo facility in our university has been shut down since August 2020. Regrettably, we can no longer use the CAM model since then.

9) In figure 5, it seems that the 0 and 5 min pictures of 0.9% NaCl are the same, and the SF scaffolds pics are also highly similar. The reviewer comes to this conclusion as the positions of the blood vessels and embryo are identical, which is nearly impossible if pictures are taken with 5 minutes difference in between (own experience). Even if there is no angioirritation from the NaCl solution and the SF scaffolds, please provide the appropriate pictures.

We changed the pictures of the treatment with NaCl and the SF scaffold in Figure 5. We also added supplementary videos showing changes taking place during the 5 minutes.

10) As of now, the manuscript is mainly descriptive as no statistical analysis is performed to determine the difference between the angiogenic response between the two groups of SF scaffolds on the different timestamps. Being merely descriptive is perfectly acceptable, but as a way to make the paper more scientifically sound, the reviewer suggests that statistical analysis is performed, and also that the authors include a ‘positive’ control within the groups to determine to what extent the hADSC are able to induce angiogenesis.

We tested for the differences by one-way ANOVA and Tukey's multiple comparison posttest as shown below. We found that the statistical test supported our claim.

E11

ANOVA

Score					
	Sum of Squares	df	Mean Square	F	Sig.
Between Groups	27.733	2	13.867	83.200	.000
Within Groups	2.000	12	.167		
Total	29.733	14			

T

Score

Tukey B ^a				
Type	N	Subset for alpha = 0.05		
		1	2	3
Filter paper	5	.00		
SF scaffold	5		2.40	
SF ADSC scaffold	5			3.20

E14

ANOVA

Score					
	Sum of Squares	df	Mean Square	F	Sig.
Between Groups	7.104	2	3.552	13.853	.001
Within Groups	3.333	13	.256		
Total	10.437	15			

]

Score

Tukey B ^{a,b}			
Type	N	Subset for alpha = 0.05	
		1	2
Filter paper	6	2.33	
SF scaffold	5		3.60
SF ADSC scaffold	5		3.80

E18

ANOVA

Score

	Sum of Squares	df	Mean Square	F	Sig.
Between Groups	7.104	2	3.552	13.853	.001
Within Groups	3.333	13	.256		
Total	10.437	15			

Score

Tukey B^{a,b}

Type	N	Subset for alpha = 0.05	
		1	2
Filter paper	6	2.33	
SF scaffold	5		3.60
SF ADSC scaffold	5		3.80

For the positive control, we cannot do it because the chick embryo facility was shut down by the university. From our results, we are sure that the SF scaffolds can promote angiogenesis. If we can have another project on this, we will definitely add VEGF or other angiogenic molecules to the SF scaffold to assess if they have additive or synergistic effect.

References

1) Reference nr 4 is incomplete

The Ref no 4 now becomes Ref no 5. We have added complete detail to it.

Language

1) In general: English is understandable but definitely needs editing as many small mistakes are noticeable in the text (see a few examples below). Also, sometimes sentences are phrased oddly (see examples below), and should be edited as well.

a. Small mistakes are noticeable throughout the text:

i. Use 'the' in front of 'chick chorioallantoic membrane' or 'CAM model' etc.

ii. Page 3:

1. 27: in THE formation

2. 50: drawn attentionS

b. Phrasing is a bit odd in several sentences throughout the whole text

i. Page 4, 3: "A great amount of work have shown the osteogenic potential of silk fibroin-based scaffolds"

ii. Page 6, 41: “Four independent assays were done to find averages.”

iii. Abstract: “In contrast, the control group could not induce angiogenesis to the same extent even as late.”

iv. Etc.

Our manuscript was edited using Grammarly.com because we need to keep the rest of our funding for the article processing charge. If there is a discounted service for us, we will be happy to use.

Reviewer 2

Majors:

1) On page 12 in line 38 – 43, the authors wrote “Moreover, there were different sizes of blood vessels inside the scaffolds. The penetration of the blood vessels was seen as early as at E11 when the SF-hADSC scaffolds were implanted.”. This information is descriptive rather than qualitative. This reviewer suggests the authors to quantify the blood vessel size and area of blood vessels per cross-sectional area of the scaffold. This is important for the study of angiogenic potential and adds up qualitative information to Figure 4.

The number pictures that we have is not enough to statistically represent the entire scaffolds. Also, the chick embryo facility in our university has been shut down since August 2020. Therefore, we cannot get more pictures for quantification.

However, we did add the quantification of the angiogenesis scores using one-way ANOVA with Tukey’s multiple comparison posttest, and found that the result is consistent with the study on the blood vessel penetration. Briefly, the SF without cells showed the angiogenic effect later than the SF with the hADSCs.

2) This reviewer would suggest the presentation of the data using mean \pm SD, instead of mean \pm SEM.

We changed from SEM to SD (Figure 2 and Table 1).

3) For the angi irritation test, only IS score of 0 and 3 has been reported. How many independent experiments have been performed? This should be mentioned.

We already stated in the HET-CAM assay subsection in the method section that “The assay was done in triplicate”. We also added (n=3...) in the figure legend.

4) Limitation of the study should be discussed. Especially, the fate of the hADSC in SF scaffolds during and upon the CAM assay was not studied.

We added another paragraph (now paragraph 6) in the discussion section that “In addition, the fate of the hADSCs in the scaffolds should be studied in the future. Our experiment provides preliminary

data that the presence of the stem cells in the SF scaffolds accelerates the angiogenesis, but cannot pinpoint whether the new vessels were formed from the hADSCs themselves or from the interaction between the hADSCs and the cells of the chick embryo.”

5) The authors should discuss why the filter paper in the CAM assay triggers embryonic death.

We added to the end of paragraph 5 in the discussion section that “Furthermore, chick embryos lack the specific immune system until E14 [44]. We believe that implanting on E8 was early enough to observe changes induced by the scaffolds. The presence of the specific immune system after E14 might explain why the embryos implanted with the filter paper died before reaching E18.”.

Minors:

1) Glucose concentration of the DMEM media should be mentioned in the cell culture section in the materials and methods, since it is critical for culture of hADSC. The authors can also provide the catalog number of the media.

We used DMEM with high glucose from Gibco. So, we added with high glucose (DMEM; Cat No: 12100046.....)

2) In the cell scanning electron microscope section in the materials and methods, please provide the settings for the imaging.

We added “The SEM was operated at the accelerating voltage of 3 keV.”

3) This reviewer would suggest to add “representative images of ...” in Figure 1B, 2B, 4 and 5.

We added “Representative” to the figure legends mentioned by the reviewer.

4) The number of experiments should be stated in Figure 2 and Table 1

In figure 2, we added “(n=4)” and “(n=3)”. In Table 1, we added “(n=5).”

5) The author discussed on collagen hydrogel and angiogenesis on Page 17 line 37-43. Recent literature suggests that “collagen microarchitecture can modulate endothelial sprouting by modulating hADSC proangiogenic capability. (Seo B et al 2020 PNAS – DOI: 10.1073/pnas.1919394117)”. In addition, proinflammatory mediators, such as IL-6, could also trigger angiogenic potential in 3D scaffolds (Sapudom J et al 2020 Bioengineering. DOI 10.3390/bioengineering7020033) (Witzel I et al 2018 Adv Healthcare Matter. DOI 10.1002/adhm.201801126). This can be added to the discussion.

We think that the first work mentioned by the reviewer can make our discussion clearer. It helps connect how SF and hADSCs interact to promote angiogenesis. So, we added “The ECM microarchitecture was shown to promote the endothelial sprouting [54].”.